# Effects of a School-Based Intervention for Preventing Substance Use among Adolescents at Risk of Academic Failure: A Pilot Study of the Reasoning and Rehabilitation V2 Program

**DOI:** 10.3390/healthcare9111488

**Published:** 2021-11-01

**Authors:** Raquel Alarcó-Rosales, Miriam Sánchez-SanSegundo, Rosario Ferrer-Cascales, Natalia Albaladejo-Blazquez, Oriol Lordan, Ana Zaragoza-Martí

**Affiliations:** 1Department of Health Psychology, Faculty of Health Science, University of Alicante, 03690 Alicante, Spain; alarco.rosales@gmail.com (R.A.-R.); rosario.ferrer@ua.es (R.F.-C.); natalia.albaladejo@ua.es (N.A.-B.); 2Department of Business Organization, Escuela Superior de Ingenierías Industrial, Aeroespacial y Audiovisual de Terrass, Universitat Politècnica de Catalunya, 08222 Terrasa, Spain; oriol.lordan@upc.edu; 3Department of Nursing, Faculty of Health Science, University of Alicante, 03690 Alicante, Spain; ana.zaragoza@ua.es

**Keywords:** reasoning and rehabilitation, substance use, intervention, adolescents, Spain

## Abstract

Tobacco, alcohol and cannabis use are important health problems that contribute greatly to causes of death in worldwide. Early onset of substance use increases rapidly during adolescence and it has been associated with a wide range of adverse events. Because substance use is associated with dramatic consequences, delaying the initiation of substance use among adolescents remains a major public priority. This study examined the effectiveness of a school-based intervention program based on the application of the Reasoning and Rehabilitation V2 (R&R2) program for preventing substance use among adolescents at risk of academic failure. A sample of 142 participants (aged 13–17 years old) who were studying alternative education provision in Spain were randomly allocated to two conditions (68 experimental group, 74 control group). A pre-test survey assessing past and current substance use was conducted prior the implementation of the program, while a post-test survey was conducted about 12 months after the pre-test. The results showed a significant effect of the R&R program in the reduction of cigarette smoking, episodes of drunkenness, alcohol consumption and cannabis use. However, for daily smoking and episodes of drunkenness such benefits showed a reduction over time. These findings offer additional evidence of the effectiveness of the Reasoning and Rehabilitation V2 program in Spanish adolescent students who are exposed to substance use and suggest areas of future research.

## 1. Introduction

Substance abuse is an important health problem that contributes greatly to causes of death worldwide. According to the World Health Organization (WHO), substance abuse is responsible for 11.8 million premature deaths each year [1]. Almost 8 million people die as a result of tobacco use or exposure, 2.9 million people die because of alcohol intake and around 600,000 people die as a result of illicit drug use [2].

Alcohol, tobacco and cannabis are the most widely used drugs among young people. According to the National Standardized Survey about drug use (ESTUDES) in Spain, around 77.9% of adolescents aged 14 to 18 years reported alcohol use at least once during their lives, with the estimated prevalence of last-year alcohol use being 75.9% of the participants. Furthermore, tobacco was the second most widespread drug among students registering that 41.3% of the students had smoked tobacco at least once in their lives and 35% of the students reported to smoke tobacco during the last year. Cannabis was the third most common drug among students aged 14 to 18 and the substance illegal with higher prevalence. Thirty-three percent of students admitted to having used cannabis at some occasion, while those who consumed in the last year accounted for 27.5% [3].Prevalence rates of these substances increase rapidly during adolescence and contribute to higher levels of use and abuse in adulthood [4]. Studies examining the pattern of initiation of drug abuse have shown that most adults who display substance abuse problems begin to use substances in early adolescence [4]. Early onset of substance use is a precursor of dependence in later life and has been associated with a wide range of adverse events including academic problems, sexual assault and aggressive behavior, as well as physical and mental health disturbances [5]. Because substance use is associated with dramatic consequences, delaying the initiation of substance use among adolescents remains a major public priority. From this perspective, the World Health Organization has emphasized the need to increase efforts to prevent the early onset of substance use worldwide [6,7].

School-based prevention programs have been recognized as one of the most effective strategies for reducing substance use among youth people [8]. A large body of research has examined the effectiveness of youth-prevention programs for substance abuse. Findings suggest that substance use prevention programs that incorporate skills-training aimed at changing attitudes, promoting social and emotional abilities, critical thinking and problem solving produce more beneficial outcomes than traditional intervention approaches focused only on changing perceptions and attitudes towards drugs [9,10]. Some examples of successful school-based programs, include the School Health and Alcohol Harm Reduction Project (SHAHRP) in Australia [11], Project ALERT [12] and ALERT Plus [13] in the United States, as well as the Unplugged program [14], which has been tested in seven European countries. These programs have shown promising results in enhancing the reduction of substance use among youth people. However, it is not clear whether these programs are effective in students at high risk of failure in school and how they achieve improvements in personal and social competences. Previous research suggests that students who are not engaged in school, who are failing academically or who feel disconnected from their schools are more likely to engage in risky behaviors including substance use and delinquency [15]. From this perspective, in the present study we implemented a school-based intervention focused on the application of the Reasoning and Rehabilitation program (R & R), a cognitive behavioral program that aims to change cognitive deficits and improve social and emotional skills in juvenile and adult populations. Evidence from previous studies have demonstrated that intervention programs that incorporate behavioral and skills training at changing attitudes and promoting social and emotional abilities during early adolescence show promising results in reducing the consumption of substances and promoting attitudes leading to the rejection of drugs [16,17,18]. It has been shown that increasing student’s awareness about the negative effects of the peer pressure that support substance use may be a way of avoiding the consumption of drugs [4].

The R&R program includes a wide range of strategies to improve problem-solving, social perspective-taking, critical reasoning, empathy, negotiation skills and values [19]. Since the original program was designed, many different versions have been developed and adapted in different countries, including North America, Canada, Australia, the UK and Spain [20]. In a previous study, we have shown that the R & R program increases levels of self-esteem, social skills, empathy and rational problem-solving in adolescent students at high-risk of failure in school. The R&R program has also demonstrated its effectiveness for reducing drug consumption in adult populations [21]. It has been also tested in youth population, specifically for those who have problems at home and school and are at risk of progressing to more severe behaviors associated with educational underachievement and deficit in prosocial and emotional competences [22].

However, the effect of this program on the reduction of drug use in young people remains unclear due to no studies to date have examined the effect of this program on substance use in adolescents. Some evidence suggests that students who fail in school and do not obtain basic studies are at greater risk of experiencing problems of social exclusion due to their difficulties in accessing to the labor market in adulthood [23].

Additionally, these students display more deficits in empathy, social and emotional skills, self-esteem and rational problem-solving [24]. Therefore, the aim of this study was to conduct a pilot study of the effectiveness of the Reasoning and Rehabilitation V2 (R & R2) program in the reduction of drug consumption in adolescent students attending alternative education provision, labeled “basic vocational training”. These programs comprise an alternative for those students who are willing to complete their secondary education but may not be suitable or able to engage or benefit within traditional schooling. This group of adolescents are specially characterized by presenting learning disabilities and being at high-risk of engaging in risky behaviors consumption of drugs, academic failure and antisocial behavior [25].

## 2. Materials and Methods

### 2.1. Sample

The initial sample was composed of 183 students at high risk of academic failure attending alternative school programs in Alicante (Spain). Students were selected from the 6 high schools covering alternative school programs. Participants ranged in age from 13 to 17 years (M = 16.04, SD = 0.83). Inclusion criteria for the students were (1) being part of an alternative education provision (2) regular attendance in the classroom (at least 80% of sessions of the program) and (3) being able to read and complete the questionnaires on their own. Students were randomly assigned to two conditions using a cluster sampling design in two stages: schools were selected by probability-to-size sampling and the random selection of a class room, with students from 13 to 17 years old attending alternative school programs.

#### Power Calculation 

Given the lack of previous studies examining the effects of treatments using the R&R V2 program for drug consumption in this population, sample size was estimated by a power calculation based on data obtained from a previous study conducted by Mitchell and Palmer [26] in youth offenders. Calculations of MANOVA global effect at 80% power and an alpha level of 0.05 suggested that 31 participants per group will be needed in order to detect this effect with a medium effect size of 0.25.

From the initial sample, 41 participants were excluded due to inclusion/exclusion criteria. The remaining 142 students were assigned to the experimental group (*n* = 68; 54.5% females; 45.5% males) and the control group (*n* = 74; 54.5% females; 45.5% males). At follow-up, 38 participants from the experimental group and 35 participants from the control group had left the intervention program and were classified as non-completers. These participants had either stopped attending school or refused to be part of the follow-up data collection. At the end of the study the sample size comprised 36 students assigned to the control group and 33 students assigned to the experimental group. An attrition analysis was conducted to compare the characteristics of the drop-outs in pre and post-intervention (Table 1). No significant differences were found.

### 2.2. Procedure

The study was approved by the Ethics Committee of the University of Alicante (UA-2015-10-13). Participants were informed about the study and the process of withdrawing and were asked to consent in order to participate. Consent was also taken from legal tutors and parents, in accordance with the Royal Decree 1720/2007 about Personal Data Protection. Participants were informed that their participation in the study was voluntary and that they could withdraw from the study with no consequences. Students’ or parents’ consent was obtained from 85% of the sample. For all students who did not return a signed form, several additional attempts were made by project staff. Parental refusal was obtained from less than 5%.

### 2.3. Experimental Design

Participants were selected from six similarly sized high schools offering alternative educational provision for students at high risk of school failure in Alicante (Spain). A total of 13 classes were identified (average class size: 11 students) and classes were then assigned to one of two experimental conditions—experimental group (EG, *n* = 6) and waiting-list control (CG, *n* = 7). Demographic data were gathered from participants, including age, sex, and nationality, academic attainment (having to repeat a course of education), economic resources, prior absenteeism at school and levels of anxiety, depression and stress. All participants completed a pre-/post-test and follow-up battery of questionnaires. For the experimental group, the pre-test battery was followed by the implementation of the R & R2 intervention program over the subsequent 12 weeks. The program was applied from January to March of the 2017 academic year. The program consisted of twelve 2-h class sessions distributed across six months. A cut-off of >10 sessions out of the 12 was used to classify students as completers (representing at least 80% attendance of the program). Pre-test data were collected during September 2018. Post-test data were collected after 12 months of follow-up. Data from baseline and follow-up were matched using a self-generated anonymous code with the purpose of linking those generated between pre-test and post-test.

### 2.4. Reasoning and Rehabilitation V2 Intervention Program

The R&R2 [27] is highly a structured twelve to fifteen 2 h sessions program focused on training cognitive, attitudinal, emotional and behavioral characteristics that are associated with negative behaviors and mental health problems in youth and in the adult population. The program is based on the evidence of more than 100 independent studies examining the effectiveness of offender rehabilitation programs [28] and it has been administrated to more than 80,000 juvenile and adult in 26 countries. The R&R program manuals have been translated into 16 languages [20]. Although the original R&R program was targeted at medium- to high-risk offenders, the R&R2 is available for lower risk offenders as well as for individuals who have not progressed toward illegal behavior. The program has been implemented in secure hospitals for mentally disordered offenders, jails and prisons, institutions for delinquent youth, probation, group homes and social service agencies [20]. It has also been developed for “adolescents and young adults with impulse control problems who lack essential constructive planning, organizational and prosocial skills and values and are engaging in various disruptive and anti-social behaviors at home, in school, at work or in community” (p. 18). The program offers a novel approach based on a body of evidence that the development of pro-social skills is associated with positive functioning [27]. The R & R program includes five modules focus on improving a set of domains including neurocognitive abilities, problem solving, emotional competences, social skills and critical reasoning [22]. The R&R program includes a wide range of strategies to improve problem-solving, social perspective-taking, critical reasoning, empathy, negotiation skills and values [19]. In order to develop such components, the program includes techniques from cognitive and social learning theories putting into practice games, practical skills and moderated discussions [26]. In this study, the intervention program was applied by qualified trainers (psychologists and educators) who had received formal training on how to conduct the program activities.

### 2.5. Outcome Assessments

Information from participants were collected using a self-completed computerized questionnaire using the National Students School-Based Drug Survey (ESTUDES) [3], a standardized national survey about drug use among adolescents in schools. Details about the ESTUDES methodology and its data can be publicly accessed [3]. The survey is based on the application of the Global School-Based Student Health Survey (GSHS) developed by the World Health Organization (WHO). The main outcomes of the study were to determine the frequency of consumption of tobacco, alcohol and cannabis. Questions about tobacco researched lifetime use, use in the past year and past month, as well as the number of cigarettes smoked during a week. Assessment of the frequency of current alcohol consumption included five response options, from “never” to “every day”. Questions on drug use and episodes of drunkenness covered lifetime, past year and past month experiences, in line with previous studies examining changes in patterns of drug consumption in adolescents in Europe [29]. A test–retest evaluation of the repeatability of the consumption of drugs was conducted before administering these questions in the main study. It was found that the recent use of cigarettes, cannabis and episodes of drunkenness showed a concordance higher than 94%.

### 2.6. Data Analysis

We conducted a descriptive analysis of the baseline characteristics of the pool intervention and control groups. A *t*-test comparison was employed to analyze possible differences in baselines between groups. Prevalence of the consumption of tobacco, alcohol and cannabis were analyzed in the past 30 days, at 6 and 12 months after intervention. In accordance with previous studies, current substance use (i.e., use in the past 30-days) is an accurate predictor of future use and subsequent escalation of the use of substances [29]. Thus, we used the following categories: (i) daily cigarette smoking; consumption of cigarettes in the last 30 days and in the last 6 months (ii) alcohol consumption in the last 30 days and in the last 6 months; (iii) any cannabis use across the life-spam. Furthermore, we also examined any episode of drunkenness to compare with previous studies. Effect size estimation was computed for each pair of variables using Cohen’s definitions (1988) [30]. MANCOVA of repeated measures of ‘time’ (pre- and post-training assessments) with ‘group’ (participants from the EG vs. participants from the CG) as a between-subject factor was performed to analyze the effectiveness of the intervention in reducing substance use. For significant results, partial eta-squared (η^2^) values were reported as a measure of the effect size [31]. Age and pre-test score age were introduced into the analyses as covariates. All statistical analyses were performed using SPSS (International Business Machines Corporation (IBM), Armonk, NY, USA), Statistics for Windows, Version 23.0, considering *p* < 0.05 to be significant.

## 3. Results

### 3.1. Baseline Sociodemographic and Clinical Characteristics of Intervention and Control Groups

Baseline analysis of the intervention and control groups in terms of the sociodemographic and clinical data are presented in Table 2. Both groups showed similar pre-treatment mean scores in sociodemographic variables except for age, where the control group were significantly younger than the experimental group. In addition, both groups reported similar levels of anxiety, depression and stress.

### 3.2. Results on Substance Use Prevalence

Changes in the prevalence of past substance use between the pool intervention and control groups at the 6 months and at 12 months follow-up are presented in Figure 1. Controlling from baseline scores, the control group displayed a significant increase in the use of tobacco, drunkenness episodes and cannabis compared to the pool intervention group. This was particularly significant in the case of cigarette smoking, of which the prevalence increased from 19.4% at post-intervention to 23.5% at 12 months follow-up in the control group. Students from the pool intervention group showed a reduction of 16.2% in the prevalence of daily cigarette smoking, 20.3% of episodes of drunkenness and 8.4% in the use of cannabis use after intervention (6 months). However, an increased risk of daily smoking and episodes of drunkenness was displayed in the treated group across 12 months of follow-up. For cannabis use, the pool intervention group showed a significant reduction of −8.4% at post-intervention and −12.4% at 12 months follow-up.

### 3.3. Effectiveness of the Reasoning and Rehabilitation V2 Program for Reducing Tobbaco, Alcohol and Cannabis Use

We analyzed the changes in intensity of past substances between the pool intervention and control groups at 6 months and 12 months of follow-up. The results of the effect of the R&R program on the intensity of tobacco, alcohol and cannabis consumption showed a significant effect of interaction group × time for the consumption of all substances over time (Table 3). When age and pre-test mean scores (T1) were introduced into the model as covariates, significant statistical effects were found between T2 and T3 for all consumed substances, showing a positive change in the treated group. Participants who received the intervention displayed a significant reduction in the intensity of consumption. This effect was significant for tobacco in the previous 6 months (F = 6.31; *p* < 0.001; η^2^ = 0.08); past 30 days smoking (F = 5.97; *p* < 0.001; η^2^ = 0.08) daily smoking (F = 5.69; *p* < 0.001; η^2^ = 0.08); alcohol use in the previous 6 months (F = 20.84; *p* < 0.001; η^2^ = 0.23); past 30 days alcohol use (F = 17.53; *p* < 0.001; η^2^ = 0.20).This was particularly relevant in cannabis use. Most students from the pool intervention group showed an occasional substance use compared with the control group in the past 6 months (F = 4.10; *p* <0.001; η^2^ = 0.05).

## 4. Discussion

This study evaluated the effectiveness of the Reasoning and Rehabilitation program V2 on the reduction of substance use in adolescent students at risk of academic failure in Spain.

Overall, the present findings show that after controlling for baseline scores, students in the pool intervention group reported a reduction of daily cigarette smoking, alcohol consumption (including any episode of drunkenness) and frequency of cannabis use, compared with the control group. Exposure to the Reasoning and Rehabilitation program was associated with an estimated reduction of around 26% in cigarette smoking from post-intervention to 12 months of follow-up and of 36% for episodes of drunkenness. Furthermore, for cannabis use, a significant reduction of 11.6% from post-intervention to 12 months of follow-up was reported during the follow-up.

These findings are in line with those found by Faggiano et al. [29], who examined the effect of the Unplugged school program, one of the most innovative school-based programs, which includes a set of life skills such as assertiveness, decision-making and coping strategies, with the purpose of delaying the onset of substance misuse among junior high school students in seven European countries.

Similar to the Unplugged program, the Reasoning and Rehabilitation program offers a wide range of cognitive, attitudinal, emotional and social skills and values based on a body of evidence that suggest that the development of pro-social skills such as self-esteem, assertiveness, self-control, empathy and problem-solving are key components to promote attitudes leading to the rejection of drugs. However, the most significant difference of our program compared with other international approaches is that we conducted our study with adolescent students with academic trajectories of failure in school. Previous research has reported that school may play an important role in the prevention of substance use, given that the initiation of substance use occurs typically during early adolescence and may predispose people to substance use later in life. In addition, research has demonstrated that students who fail academically or feel discomforted from their school are more susceptible to engaging in substance use [4]. From this perspective, schools and environmental contexts may play an important role in the prevention of substance use. It has been reported that school-based interventions represent a promising approach to enhancing success in school and increasing interpersonal skills in adulthood [18]. A recent review of interventions for tobacco use, alcohol use, drug use and combined substance abuse among adolescent students demonstrated that school programs that are focused on a combination of a social competence and social influence approach have a more positive impact in preventing substance use, particularly when programs include antidrug information combined with refusal skills, self-management and social skills training [32].

The results of our study support these prior findings, showing that those students who received the intervention program displayed a significant reduction in the intensity of drugs consumption during the follow-up. These findings were particularly relevant for cannabis use. However, from post-intervention to 12 months of follow-up an increase in substance use was observed for daily smoking and episodes of drunkenness. Prior studies have reported that intervention programs are able to produce positive changes in substance use and other psychosocial outcomes; however, such benefits tend to diminish over time [33]. Our findings are in line with previous findings, suggesting that although intervention programs may represent a promising approach to reducing substance use and increasing interpersonal skills, it is necessary that intervention can be maintained across time as part of a continuum of care and monitoring, particularly in school. Examining factors related to relapse can provide an empirical basis for identifying effective intervention approaches [34]. The present findings also suggest that efforts should be concentrated on the early identification and routine monitoring of adolescent substance use. Given that most unhealthy behaviors such as smoking, drinking and illicit drug use often begin in early adolescence, it is essential that intervention programs can be incorporated with school curriculum programs. Evidence accumulated to date has demonstrated that schools and communities can play a protective role by taking steps to engage students in order to avoid substance use and other unhealthy problems by teaching students specific skills for resisting pressure to smoke drink or use drugs [4]. Thus, prevention initiatives should be designed with the goal of increasing adolescents’ awareness of the effects of substance use over the long term and teaching them to recognize situations in which they are likely to experience pressure to use drugs. Some prior research has shown that the most effective programs are those focused on training skills and interactive programs that show a comprehensive message and are implemented over multiple years [4].

There are several limitations in the current study that suggest areas for future research. Firstly, the current study was conducted with students attending alternative school programs, designed for those students at high risk of failure in school. Therefore, researchers and clinicians must use caution when generalizing these findings to other students. Future research should explore the impact of R&R2 on adolescent populations attending traditional academic curricula in mainstream school settings. Secondly, outcome measures were gathered using self-reported data and it is possible that some adolescents may have underestimated or exaggerated their responses. Third, we examined a small sample of adolescent students, which limited the statistical power of the data. From the initial sample we observed a drop-out around 50% in both control and experimental groups. Similar results have been found in previous longitudinal intervention studies in high risk adult population. For example, a preliminary study of the R&R in adult offenders in drug treatment in the United State found a drop-out rate of 44.6%. Additionally, evidence of high rates of treatment drop-out has been found in other studies. Ashford et al. [35] reported a drop-out rate of 51% in mentally disordered offenders in community, suggesting that adherence to treatment may be more difficult to achieve in community settings. Furthermore, in the study conducted by Young et al. [36] in mentally ill offenders attending the R & R2M program, from initial sample of 47 patients, 28% failed to start treatment and 25% drop-out in the follow-up. These results suggest the importance to examine the factors predicting drop-out for the Reasoning and Rehabilitation program, particularly in community setting where rates of treatment non completion are higher [37].

## 5. Conclusions

The current study extends the previous evidence of the effectiveness of the R&R program and supports the effect that school-based interventions may have on substance use in adolescent students. Intervention programs such as the R&RV2 program may represent a promising approach to reducing substance use and increasing interpersonal skills. However, it is necessary that intervention can be maintained over time as part of a continuum of care and monitoring. As schools provide more opportunities for conducting intervention programs, it is essential that intervention programs can be incorporated within school curriculum programs.

## Figures and Tables

**Figure 1 healthcare-09-01488-f001:**
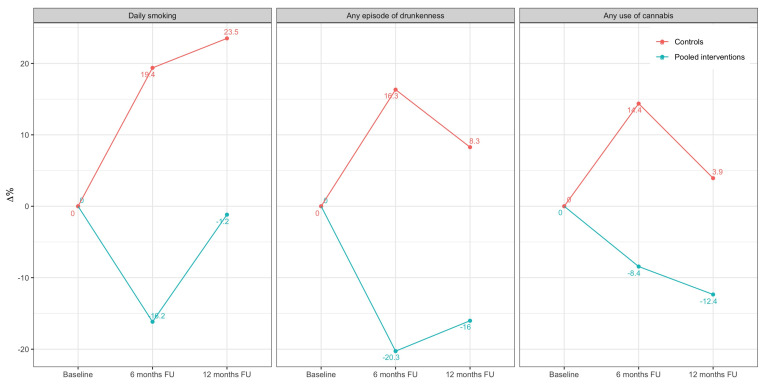
Changes in prevalence of past substance use between experimental and control groups at baseline, and at the 6-month and at the 12-month follow-up.

**Table 1 healthcare-09-01488-t001:** Attrition analysis of the number of drop-outs at baseline and follow-up.

	Baseline (T1)	Follow-Up (T3)	95% CI	SD	*p* Value
Alcohol	1.33	1.43	−0.67 to 0.46	0.28	0.71
Tobacco	0.77	0.75	−0.32 to 0.36	0.17	0.90
Cannabis	1.18	0.91	−0.33 to 0.86	0.30	0.38

Note: CI = confidence interval; SD = standard deviation.

**Table 2 healthcare-09-01488-t002:** Baseline sociodemographic and clinical characteristics of experimental and control groups in Time 1 (T1).

	Control Group (*n* = 74)		Pool Intervention (*n* = 68)			
	Men *n* (%)	Women *n* (%)	Men *n* (%)	Women *n* (%)	χ^2^	*p*
Sex	48 (64.9)	26 (35.1)	52 (76.5)	16 (23.5)	2.29	0.130
	Spanish *n* (%)	Other *n* (%)	Spanish *n* (%)	Other *n* (%)	χ^2^	*p*
Nationality	62 (83.8)	12 (16.2)	58 (85.3)	10 (14.7)	0.06	0.804
	Yes *n* (%)	No *n* (%)	Yes *n* (%)	No *n* (%)	χ^2^	*p*
	Mean (SD)		Mean (SD)		t	*p*
Age *	15.93 (0.60)		16.15 (0.63)		−2.07	0.040
Economic resources	17.97 (20.27)		12.29 (15.75)		1.83	0.066
Anxiety	3.39 (4.54)		3.69 (4.19)		−0.33	0.739
Depression	4.13 (5.16)		3.98 (4.37)		0.15	0.879
Stress	3.80 (4.09)		4.22 (4.69)		−0.45	0.654

Note: * *p* < 0.05.

**Table 3 healthcare-09-01488-t003:** Differences between experimental group (EG) and control group (CG) adolescents on post-intervention and follow-up.

Outcome	Group	Pre-Training (T1)		Post-Intervention 6 Months (T2)		Follow-Up 12 Months (T3)		Group Effect		
		Mean	SD	Mean	SD	Mean	SD	*F*	Direction	ŋ^2^
Tobacco (daily)	Experimental	1.65	4.24	0.13	0.79	1.58	4.22	5.69 **	EC < CG	0.08
	Control	2.15	3.89	3.12	4.40	2.61	3.86			
Tobacco (30 days)	Experimental	0.55	1.06	0.23	0.53	0.55	1.09	5.97 *	EC < CG	0.08
	Control	1.00	1.32	1.33	1.43	1.39	1.48			
Tobacco (6 months)	Experimental	0.75	1.35	0.40	0.87	0.28	0.45	6.31 **	EC < CG	0.08
	Control	1.33	1.78	1.70	2.27	0.48	0.51			
Alcohol (30 days)	Experimental	1.50	2.03	0.08	0.27	1.63	1.21	17.53 ***	EC < CG	0.20
	Control	2.18	2.16	2.67	2.13	2.94	2.21			
Alcohol (6 months)	Experimental	3.30	2.96	0.47	0.64	2.48	1.63	20.84 ***	EC < CG	0.23
	Control	3.33	2.89	3.79	2.63	4.12	2.67			
Cannabis (6 months)	Experimental	2.01	2.79	1.65	2.61	0.98	2.04	4.10 *	EC < CG	0.05
	Control	1.69	2.62	2.21	2.68	1.79	2.65			

Note: *** *p* < 0.001; ** *p* < 0.01; * *p* < 0.05; EG = experimental group; CG = control group.

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
