# Peer review of "Effects of a School-Based Intervention for Preventing Substance Use among Adolescents at Risk of Academic Failure: A Pilot Study of the Reasoning and Rehabilitation V2 Program"

_healthcare, 2021, doi:10.3390/healthcare9111488_

Round 1

Reviewer 1 Report

Thank you for the opportunity to read an interesting text related to the prevention of health behaviour. The authors rightly point out that easily accessible psychoactive substances are the cause of numerous health and social problems. Prevention among adolescents is not an easy task. For many years, behavioural addictions have been a challenge for various groups of educators, parents (primary environment) or the strongly developing concept of peer education. Taking into account the importance of the subject matter the reviewed text should be rated very highly. However, I have a few comments, which I present below, because I have the impression that the text could still be improved a little.

  1. In the abstract it is useful to add information about the research sample (number of people), the mediating variable (so what made up the intervention programme). 
  2. The country in which the research was conducted can be added in the keywords.
  3. In the introduction, it is useful to show current statistics on the consumption of psychoactive drugs by young people in Spain. This is one of the arguments why this type of activity should be undertaken. You can also show data related to trends in the country where the research was conducted.
  4. The inclusion of pilot studies focused on one group is not very understandable. Why were students at high risk of failure in school linked to a test of a prevention programme? This needs stronger justification. Hard data from the country where the study was conducted is needed in this regard. One might get the impression that only this group needs preventive support, which is not entirely true.
  5. In the criteria for inclusion of the respondents, the authors use the term - alternative educational program . However, it is not clear what this means.
  6. Regular attendance in the last 3 months was also a selection criterion. I will say honestly that this criterion has no justification. It is not understandable to me. It is worth developing this thread.
  7. The procedure of the pedagogical experiment (the article can be attributed as such) should be presented in a diagram in order to increase the clarity of how the respondents were selected and how the research was conducted.
  8. The R&R programme is poorly described in the article. How long was one session, what teaching methods, forms and means were used, who taught, what were the operational objectives? A detailed description of the programme components is missing from the text.
  9. The differences in project effectiveness are very apparent. In particular, the maintained downward trend for cannabis is impressive. I congratulate you on preparing an effective programme and encourage you to consider what you need to do to maintain the durability of the effects beyond the six-month period.

Overall, congratulations on producing an interesting article. Making changes will perhaps make it more readable for international readers unfamiliar with Spanish conditions. I would be delighted to see an updated version of the study.

Author Response

1. In the abstract it is useful to add information about the research sample (number of people), the mediating variable (so what made up the intervention programme). 

 We have highlighted in yellow the requested information within the abstract section.

2. The country in which the research was conducted can be added in the keywords.

 Thank you! It has alredy been added.

3. In the introduction, it is useful to show current statistics on the consumption of psychoactive drugs by young people in Spain. This is one of the arguments why this type of activity should be undertaken. You can also show data related to trends in the country where the research was conducted.

We have included the following information in the introduction section:

Alcohol, tobacco and cannabis are the most widely used drugs among young people. According to the National Standardized Survey about drug use [ESTUDES] in Spain, around 77.9% of adolescents aged 14 to 18 years reported alcohol use at least once during their lives, with the estimated prevalence of last-year alcohol use being 75,9% of the participants. Furthermore, tobacco was the second most widespread drug among students registering that 41.3% of the students had smoked tobacco at least once in their lives and 35% of the students reported to smoke tobacco during the last year. Cannabis was the third most common drug among students aged 14 to 18 and the substance illegal with higher prevalence. 33.0% of students admitted to have used cannabis in some occasion, while those who consumed in the last year account for 27.5%.

4. The inclusion of pilot studies focused on one group is not very understandable. Why were students at high risk of failure in school linked to a test of a prevention programme? This needs stronger justification. Hard data from the country where the study was conducted is needed in this regard. One might get the impression that only this group needs preventive support, which is not entirely true.

Thank you for your suggestion! We have included a paragraph at the end of the section to justify the inclusion of this population.

5. In the criteria for inclusion of the respondents, the authors use the term - alternative educational program . However, it is not clear what this means.

 We have answered this question in the last comment.

Alternative education provision, labelled “basic vocational training”. These programmes comprise an alternative for those students who are willing to complete their secondary education but may not be suitable or able to engage or benefit within traditional schooling. This group of adolescents are specially characterised by presenting learning disabilities and being in an at risk or in a situation of social exclusion and thus they are vulnerable in problematic consumption of drugs, academic failure and antisocial behaviour.

6. Regular attendance in the last 3 months was also a selection criterion. I will say honestly that this criterion has no justification. It is not understandable to me. It is worth developing this thread.

We have clarified the criterion into the paragraph. The selection criterion was to attend at least the 80% of the sessions of the program.

7. The procedure of the pedagogical experiment (the article can be attributed as such) should be presented in a diagram in order to increase the clarity of how the respondents were selected and how the research was conducted.

This study was part of another published paper whichin the flow diagram of participants recruitment was published. This paper can be seen in the following reference:

Sánchez-SanSegundo, M., Ferrer-Cascales, R., Albaladejo-Blazquez, N., Alarcó-Rosales, R., Bowes, N., & Ruiz-Robledillo, N. (2020). Effectiveness of the Reasoning and Rehabilitation V2 Programme for Improving Personal and Social Skills in Spanish Adolescent Students. International journal of environmental research and public health17(9), 3040.

8. The R&R programme is poorly described in the article. How long was one session, what teaching methods, forms and means were used, who taught, what were the operational objectives? A detailed description of the programme components is missing from the text.

 We have developed more information about the programme.

9. The differences in project effectiveness are very apparent. In particular, the maintained downward trend for cannabis is impressive. I congratulate you on preparing an effective programme and encourage you to consider what you need to do to maintain the durability of the effects beyond the six-month period.

We have reported how educational interventions based on school programs to prevent the consumption of drugs can create an environment in which students are able to refuse risky behaviors. However, given that participants of our study, are cognitively immature due to their age it is necessary that students receive reinforcements to maintain the results of the interventions in long-term. As we have demostrated, after follow-up the effect of intervention down, so it is neccesary the use of reinforcements across time.

Reviewer 2 Report

This paper examines the effects of a school-based program aims at decreasing substance use among adolescents at risk of academic failure. Despite there are already many school-based programs that have shown some effectiveness in preventing substance use among adolescents, the study is original in that it focuses on a specific subgroup of adolescents at risk of academic failure. I think the topic is very relevant and original. However, I believe that some improvements are needed before consideration for publication.

  • A main general comment is that the authors failed in explaining why this program should work specifically with this type of population. It is not even clear if the program was developed for the general population of adolescents and then tested in particular in this subgroup. The authors stated that they have already tested the effects of the program on many skills in a population of high-risk students (p.2, l. 73-76). Was this the population the target from the beginning or is it the population with which the program is effective? Please clarify this point. It would be also good to have a better description of the program and of its program theory instead of referring to another paper (p.3l, lines130-140).  How is the program supposed to decrease substance use in the specific subgroup?

Other more specific comments are in the following:

Introduction:

The introduction is generally well written and easy to follow.

Methods:

  • Please clarify the meaning of “alternative educational program”;
  • The authors stated that a detailed description of the procedure has been published elsewhere, but I think it would be good to provide some more information about the drops out. For instance, are the reason for dropping out different between the treatment and the control group?
  • As I wrote above, a better description of the program theory would help to understand why this program should work with this subgroup of adolescents;
  • I think that it would help to get a list of all the questions that have being used with a clear description of the response range for each and other indicators of reliability when appropriate;
  • It is not clear to me how the authors could run a MANCOVA analyses with such low power. Why did they not consider the ANCOVA instead?
  • The authors stated that the effect sizes were computed using Cohen definition. However, they did not use the Cohen coefficient, so the sentence might be misleading for the reader.

Results:

  • The authors presented the baseline sociodemographic and clinical characteristics of the treatment and control group. However, they should compare also the baseline means for the dependent variables to show whether those two groups were similar at baseline.
  • It would be good to add an attrition analysis to understand whether the drops outs were different in the two groups (for instance, more smokers dropped out in the control group compared to the treatment group).

Conclusions:

 In general, the conclusions are well written.

Author Response

A main general comment is that the authors failed in explaining why this program should work specifically with this type of population. It is not even clear if the program was developed for the general population of adolescents and then tested in particular in this subgroup. The authors stated that they have already tested the effects of the program on many skills in a population of high-risk students (p.2, l. 73-76).

The R&R2 programme has been tested in adult and youth population. In adolescents, the programme was specifically designed for those who have problems at home and school and are at risk of progressing to more severe behaviours associated with educational underachivement and deficit in prosocial and emotional compentences. Therefore, the present study we examined the effect of this program in adolecents that are vulnerable in problematic consumption of drugs, academic failure and antisocial behaviour. To our knowledge, the effect on this specific population has not been investigated to date.

Was this the population the target from the beginning or is it the population with which the program is effective? Please clarify this point.

As we suggested in the above question, the programme was designed for adolescents at risk of progressing to more severe behaviours associated with educational underachivement and deficit in prosocial and emotional compentences. After reviewing the literatura, we contacted with the original author of the programme who suggested the applicability of the programme for this specific subgroup of vulnerable students.

It would be also good to have a better description of the program and of its program theory instead of referring to another paper (p.3l, lines130-140).  

Thank you for your suggestion. We have included additional information within the subparagraph 2.3. R&R2 Intervention Program.

How is the program supposed to decrease substance use in the specific subgroup?

Evidence from previous studies have demonstratred that intervention programs that incorporate behavioural and skills training at changing attitudes and promoting social and emotional abilities during early adolescence show promising results in reducing the comsumption of sustances and promoting attitudes leading to the rejection of drugs. This is particularly important in those students with deficit in social competences, given that they are more vulnerable to peer pressure.  So, we expected to find out that a highly structured program such as the R&R might have a significant effect in a small group of participants.

Methods:

Please clarify the meaning of “alternative educational program”;

We have replaced within the manucript the term with “alternative education provisión”. This information has been included at the end of the introduction: “These programmes comprise an alternative for those students who are willing to complete their secondary education but may not be suitable or able to engage or benefit within traditional schooling. This group of adolescents are specially characterised by presenting learning disabilities and being at high-risk of engaging in risky behaviours including problematic consumption of drugs, academic failure and antisocial behavior.

The authors stated that a detailed description of the procedure has been published elsewhere, but I think it would be good to provide some more information about the drops out. For instance, are the reason for dropping out different between the treatment and the control group?

We found the same reasons of drops out in both groups and the loss of the participants were similar. Reasons included stop attending school or refusing to be part of the different stages of the program including the assessment of questionnaires. Non additional reasons were reported by participants.

As I wrote above, a better description of the program theory would help to understand why this program should work with this subgroup of adolescents.

The R&R program is based on the evidencie of hundreds of evaluation of rehabilitation program in different populations (Andrews & Bonta, 2010).

Most effective rehabilitation programs are based on a cognitive/behavioral model (McGuire, 2002).

One of the earliest cognitive-behavioral programs is the Reasoning and Rehabilitation (R&R) program (Ross, Fabiano, & Ross, 1986). R&R teaches offenders cognitive, emotional and social skills and values that are required for pro-social competence. It trains in skills and values that enable them to withstand environmental and personal factors that engender negative behavior. It has been delivered to more than 80,000 juvenile and adult in 26 countries. The R&R program manuals have been translated into 16 languages. It has been implemented in jails and prisons, secure hospitals for mentally disordered offenders, institutions for delinquent youth, probation, group homes, social service agencies for at-risk youth, and community schools.

Based on research published since the original program was created in 1986, a number of new and shorter versions of the original program have been developed to target specific groups of offenders and antisocial individuals: R&R2 for Youth (Ross & Hilborn, 2004); R&R2 for Adults (Ross, Hilborn, & Liddle, 2007); R&R2 for Girls & Young Women (Ross, Gailey, Cooper, & Hilborn, 2007); R&R2 for Families & Support Persons (Ross & Hilborn, 2008b); R&R2 for Youths & Adults With ADHD (R&R2ADHD, Young & Ross, 2007a); R&R2 for Youths & Adults With Mental Health Problems (R&R2MHP, Young & Ross, 2007b).

From this perspective and given that the effectiveness of drugs consumption in adults has been evaluated, we decided to conduct the program in youth at high-risk of vulnerability.

Antonowicz, D. H. & Parker, J. (2013). Reducing Recidivism Evidence from 26 Years of International Evaluations of Reasoning & Rehabilitation Programs. Wilfrid Laurier University.

Martín-Caballero, A. R.; Bethencourt Pérez, J. M.; García Medina, P.; Fernández Valdés, A., & Ramírez Santana, G. M. (2009). Valoration and training in social competence in drug addicts: A specific implementation with users of the Day Center "Cercado del Marqués". Tenerife, Canary Islands: Unpublished report.

It is not clear to me how the authors could run a MANCOVA analyses with such low power. Why did they not consider the ANCOVA instead?

While the ANCOVA analysis is used to determine whether or not there is a statistically significant difference between the means of three or more independent groups, including one or more covariates, in our case we decided to go with MANCOVA given we had multiple response variables (tobacco, alcohol and cannabis) and more than one covariables. We aware that our population is relatively small given we conducted a pilot testing of the R&R programme. Similar to our study there are some papers using MANCOVA analyses:

For example: Keshavarzi, S., Fathi Azar, E., Mirnasab, M. M., & Badri Gargari, R. (2016). Effects of a Transactional Analysis Program on Adolescents’ Emotion Regulation. Inter J Psycho Stud8(4), 51-60.

The authors stated that the effect sizes were computed using Cohen definition. However, they did not use the Cohen coefficient, so the sentence might be misleading for the reader.

 We agree. It has been removed. Thank you

Results:

The authors presented the baseline sociodemographic and clinical characteristics of the treatment and control group. However, they should compare also the baseline means for the dependent variables to show whether those two groups were similar at baseline.

The pre-training means can be seen in table 2. Participants were randomised assignated to two conditions, therefore analysis were run comparing Time 2 and Time 3 after controlling for Time 1. Also, baseline difference in age was controlled as covariable.

It would be good to add an attrition analysis to understand whether the drops outs were different in the two groups (for instance, more smokers dropped out in the control group compared to the treatment group).

Thank you for your observation. We considered to do an attrition analysis but our data wasn't adequate for it. In most patients, we have only the pre-test questionnaire given that some patients refuse answer the questionnaire after intervention program, other were classified as non-completers (a cut-off of >10 sessions out of the 12 was used to classify students as completers -representing at least 80% attendance of the program), and other leave school so we have missing data. With this, we have drop-outs for multiple reasons that doesn't allow us to determine if the treatment or the consumption has any impact in drop-out.

Round 2

Reviewer 1 Report

The authors have responded in detail to all review comments. As it stands, the text meets the acceptance criteria. The study is valuable first of all from the perspective of health behaviour prevention. The themes taken up in the text are universal for many countries. The study may turn out to be interesting for other researchers, school pedagogues and educational politicians who are looking for effective forms of first and second-line prophylaxis. I recommend the text for publication.

Author Response

Thank you very much for your help and your value comments which have improved the manuscript. 

Reviewer 2 Report

The paper improved after the revisions the authors have made. However, I do not think that the authors did answer to all my concerns.

I do not think that the authors can just answer to my doubts without making the points clearer in the text of the manuscript because other readers might have the same doubts. So I invite the authors  to integrate their answers in the text too, as they were often points that clarified parts unclear of the text.

Moreover, I would like to add two more points:

  • I understand that the best model to be performed is a MANOVA but I do not think that the authors have enough power to use that method. Please provide a sample power calculation to ensure that you can use this analysis, or a better methodological reference that confirm this sample is enough to run the analysis.
  • An attrition analysis is done comparing the scores at baseline between those that take part at both waves (pre and post) and those that participate to only the first wave (pre test). Therefore, the authors can perform the analysis with the data they have.

Author Response

I do not think that the authors can just answer to my doubts without making the points clearer in the text of the manuscript because other readers might have the same doubts. So I invite the authors  to integrate their answers in the text too, as they were often points that clarified parts unclear of the text.

Thank you very much for all your suggestions and value comments. All the answers have already been included into the manuscript. We think that all your suggestions have contributed to increase the quality of the manuscript and the interest for readers. 

1. I understand that the best model to be performed is a MANOVA but I do not think that the authors have enough power to use that method. Please provide a sample power calculation to ensure that you can use this analysis, or a better methodological reference that confirm this sample is enough to run the analysis.

Thank you very much!

We have conducted the power calculation using gpower8 programme. Results suggest an adequate power estimation.

“Given the lack of previous studies examining the effects of treatments using the R&R V2 program for drug consumption in this population, sample size was estimated by a power calculation based on data obtained from a previous study conducted by Mitchell & Palmer (2004) in youth offenders. Calculations of MANOVA global effect at 80% power and an alpha level of 0.05 suggested that 31 participants per group will be needed in order to detect this effect with a medium effect size of 0.25”.

2. An attrition analysis is done comparing the scores at baseline between those that take part at both waves (pre and post) and those that participate to only the first wave (pre test). Therefore, the authors can perform the analysis with the data they have.

According to the reviewer suggestion, we have conducted an attrition analysis by comparing the characteristics of the drop-outs in pre and post-intervention. We have included the table 1 into the text.

Round 3

Reviewer 2 Report

The authors did a great job in addressing all the reviewer´s comments and I believe the manuscript improved notably it is ready to get published.